# Towards Artist-Like Painting Agents with Multi-Granularity Semantic Alignment

## ABSTRACT

Mainstream painting agents based on stroke-based rendering (SBR) attempt to translate visual appearance into a sequence of vectorized painting-style strokes. Lacking a direct mapping (and consequently the differentiable ability) between pixel domain and stroke parameter searching space, these methods often yield non-realistic/artist-incompatible stroke decompositions, hindering its further application in high quality art generation. To explicitly address this issue, we propose a novel SBR based image-to-painting framework which aligns with artistic oil painting behaviors/techniques. In the heart is a semantic content stratification module which decomposes images into hierarchical painting regions encapsulated with semantics, according to which a coarse-to-fine strategy is developed to first fill-in the abstract structure of the painting with coarse brushstrokes; and then depict the detailed texture portrayal with parallel-run localized multi-scale stroke search. In the meantime, we also propose a novel method that integrates SBR frameworks into a simulation-based interactive painting system for stroke quality assessment. Extensive experimental results on a wide range of images show that our method not only achieves high fidelity and artist-like painting rendering effect with a reduced number of strokes, but also exhibits greater stroke quality over prior methods.

## CCS CONCEPTS

• **Do Not Use This Code** → **Generate the Correct Terms for Your Paper**; *Generate the Correct Terms for Your Paper*; Generate the Correct Terms for Your Paper; Generate the Correct Terms for Your Paper.

## KEYWORDS

Painting Agent, Stroke-based Rendering, Semantic Stratification

## 1 INTRODUCTION

With the emergence of fluid-simulation in computer graphics [2] and AIGC [1, 18], automatic painting agents have aroused great interest among many professional artists for its flexibility and application potential in digital art/design creation, art education, and interactive art, etc. Current methods are mainly based on stroke-based rendering (SBR) [8, 9, 14, 16, 19, 20, 25], which decomposes a photo-realistic image into a sequence of brushstrokes (in the format of vectorized representation) and then draws in order these

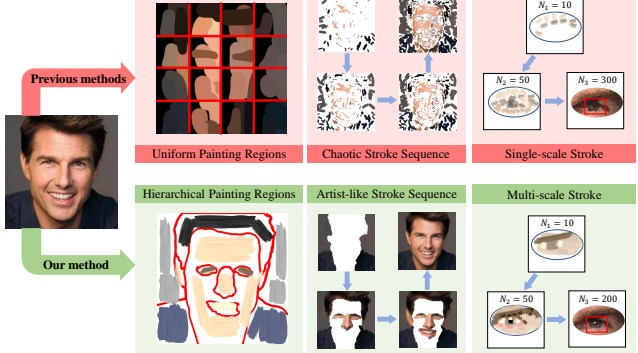

**Figure 1: Illustration of our motivation. Current methods in stroke-based rendering (SBR) fail to align with realistic human painting techniques, which can be attributed to three main issues: 1) Uniform Painting Regions [8, 9, 14] that ignore the hierarchical structure of images; 2) Chaotic Stroke Sequence [20] that is inconsistent with the artistic progressive painting habits; and 3) Single-granularity Strokes [9, 25] that accumulate in local patterns of paintings. To address above issues, this paper is motivated to develop a novel SBR method which aligns with artistic painting techniques by decomposing the image into hierarchical painting regions and generating strokes in a coarse-to-fine fashion that progresses from background to foreground, from abstract to concrete and from coarse granularity to fine granularity.**

discrete strokes on the canvas through template-based [24], neural-based [8, 9, 19] or simulation-based [2] stroke rendering techniques.

While a human artist typically plan his/her painting strokes with a strategy that progresses from abstract (overall structure fill-in) to concrete (local details highlighting), and from coarse-grained to fine-grained semantics, current stroke planning algorithms, however, mainly rely on pixel-level rendering loss based reinforcement learning, completely ignoring this important hierarchical semantic structure in outputting their stroke sequences. In other words, no artist knowledge/experience is incorporated, which apparently leads to the following serious issues: On the one hand, as the relationship between **brushstroke granularity** and the coarse-to-fine content structural semantics is NOT explicitly modelled, most methods tend to stack a large number of semantic-agnostic fine-grained strokes for reconstructing the highly textured regions [9, 20, 25], leading to great redundancy/complexity in the resulting stroke sequence. For example, a $512 \times 512$-sized image often requires more than ten thousand strokes.

On the other hand, as the optimization objective is only to recursively build the pixel gap between the ground-truth image and the rendered one, with NO regard to whether the generated stroke

is compatible with real artist's stroke characteristics like stroke ordering, the resulting rendered painting often presents non-artist style, which forbids real applications.

Pioneer works such as Intelli-Paint [19] attempt to plan the painting process in an artist-like fashion, but it relies solely on object detection for basic background-to-foreground separation and the planned strokes do not explicitly align with the coarse-to-fine image structure. Other algorithms tend to divide images into discrete/regular square regions and paint each square region independently for efficiency improvement. For instance, PaintTransformer [14] decomposes images into multi-scale patches to generate corresponding brushstrokes, which results in a pronounced visual fragmentation. DPPR [8] employs reinforcement learning to dynamically identify painting regions within an image to avoid visual fragmentation. However, these painting regions remain regular squares, thus inhibiting precise alignment with the underlying structure/semantics of the image content.

In pursuit of artistic-like painting agents and addressing the aforementioned limitations, we propose a novel image2stroke framework which enforces the generated sequence of strokes to align with artistic oil painting styles. Specifically, our framework generates a hierarchy of image painting regions/semantic segmentations through a semantic content stratification module, with the help of off-the-shelf image parsing big models such as Depth-Anything [23] and Segment-Anything [10], based on which a structure-aligned coarse-to-fine stroke ordering and parameter adaptation scheme is developed, for generating artistic-compatible stroke sequences. Specifically, with the above painting region stratification, our method generates strokes imitating the artistic painting style in a two-stage fashion. We first utilize coarse brushstrokes to fill in the abstract layout/structure of the input image content by employing the differentiable stroke renderer to optimize the stroke parameters in parallel, thereby achieving a rough yet very efficient semantic structural alignment. Second, a divide-and-conquer strategy is adopted for efficient multi-scale brushstroke search within each distinct region for a more detailed and textured portrayal in a progressive and coarse-to-fine paradigm. To mitigate boundary inconsistency artifacts between regions caused by localized stroke search, we further develop an edge compensation module based on the image gradient field.

Extensive comparisons on rendering quality with state-of-the-art methods show that our method achieves high-fidelity stroke-based rendering with a reduced number of brushstrokes. Specifically, we can achieve better visual rendering results than previous methods with a **20%** reduction in the number of strokes. When using the same amount of strokes, our method can depict significantly more realistic and finer image textures. Compared to previous work, our painting areas are characterized by the diverse shapes encapsulated with content semantics and hierarchically structural ordering, *i.e.*, from object level to local part level, evolving from intricate textures to smoother textures, thus yielding artistic-like painting styles. In addition, we develop an automatic painting agent based on physical fluid simulation [11], which uses the generated stroke sequences to automatically drive physical brushes and paint on a canvas, allowing us to directly assess the practicality/artistic quality of different brushstroke planning algorithms. The experiments demonstrate that our algorithm performs the best among the state-of-the-art.

## 2 RELATED WORK

Traditional SBR algorithms include greedy strategies [4, 6, 13], energy function optimization algorithms [7, 21], and user-guided semi-automatic methods [5]. These algorithms all utilize stroke primitives based on mathematical models, such as cubic B-spline [6], with constant texture within a stroke, which are unable to display the realistic details. Im2Oil [20] adopts a texture-encapsulated meta brushstroke [24] as painting primitive and employs adaptive sampling for stroke parameters search, which produces compelling painting textures. However, its stroke sequence is unrelated to image semantics, leading to the painting process artist-incompatible and short of practical significance.

With the popularity of deep learning technology, SBR algorithms based on neural networks raise widespread attention. [9] is a pioneering work that incorporates reinforcement learning (RL) framework [12] for realistic painting, which provides satisfying results for high-resolution scene paintings, although with noticeable artifacts at the fixed grid edges. To mimic human painting habits, Intelli-Paint [19] trains the RL model to output strokes within the special semantic region, achieving a painting sequence that progresses from background to foreground. But it relies solely on object detection for basic background-to-foreground separation and generates strokes in block-style painting region, which can not explicitly align with finer-grained image semantics.

To enhance efficiency for generating stroke parameters, Paint-Transformer [14] utilizes a self-supervised sequence-to-sequence model to generate strokes in a feed-forward manner. Zou *et al.* [25] propose a neural stroke renderer to optimize stroke parameters directly through gradient descent. However, their results are unsatisfactory, primarily due to regional voids and lack of fine-grained strokes. DPPR [8] proposes to train a RL model to dynamically predict the next painting region on the current canvas, significantly reducing the boundary artifacts caused by uniformly divided painting regions. However, DPPR requires truncating the stroke parameters within the corresponding rectangular areas for rendering, which does not align with the techniques of realistic painting.

Different from above methods, we propose a novel SBR algorithm with multi-granularity semantic alignment, which aligns the stroke generation process with artistic oil painting styles, particularly tailored for artist-like painting agents.

## 3 METHODOLOGY

As mentioned above, our novel framework contains 1) a semantic content stratification module, 2) a two-stage artist-like stroke generation/planning module along with an edge compensation module, as illustrated in Fig. 2, for the purpose of aligning artist's knowledge towards image-based efficient and high-fidelity vectorized brushstroke sequence recreation (in the style of oil painting). The overall pipeline is elaborated as follows.

### 3.1 Semantic Content Stratification Module

Previous methods divides image into regular image grids either in fixed [14] or dynamic (run-time) identified [8, 19] schemes, in order to generate brushstrokes in seperate painting regions independently. These practices seriously violate the artistic painting process: 1) boundary inconsistency/artefacts between grids are induced; and 2) the ordering and granularity of the generated brushstrokes are

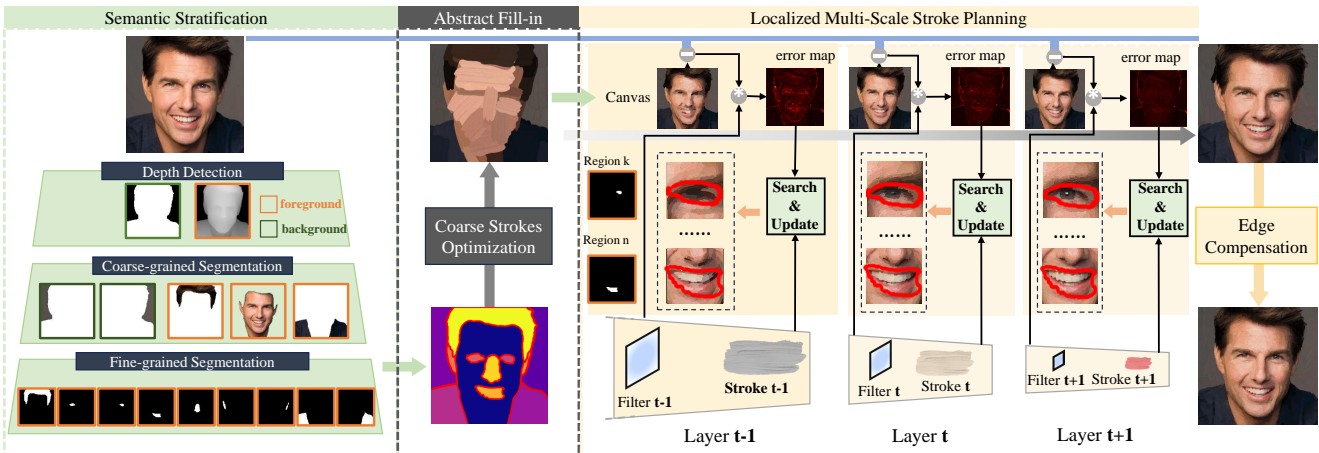

Figure 2: Overview of our framework. Our framework contains a semantic content stratification module, a two-stage artist-like stroke generation module and an edge compensation module. Our method starts from decomposing the input image into hierarchical semantic painting regions. Then a two-stage stroke generation method is utilized to firstly fill in the abstract structure of the image through optimizing a small number of coarse-grained strokes and then perform parallel localized multi-scale stroke search within each painting region to depict fine-grained image textures. Finally, we refine the boundaries among painting regions to generate better visual effects.

NOT compatible with a typical artist's behaviour. More specific, an oil painting artist usually attempts to use coarse-grained strokes to fill the rough structure of the entire scene such as the background layout, sky, and ground, etc., and then use median size strokes to outline major objects in the scene such as human body, face, etc., and finally use fine-grained strokes to depict important local object parts such as eyes, mouths, and fine textures, etc.

The above observation motivates us to propose the following semantic content stratification module to divide the image into a sequence of painting regions that not only to mimic the artistic painting ordering but also to align with the hierarchical/multi-granularity semantic information of the image content, e.g., background → human/face → eye/mouth, so that following this hierarchical/tree structure, strokes with different granularities could be planned accordingly for both efficiency and painting fidelity boosting.

For the input image $I \in \mathbb{R}^{W \times H \times 3}$, we separate it into background (with relatively smoother textures) and foreground (containing richer semantics and textures) areas. For such a purpose, we utilize Depth Anything Model (DAM) [23] to obtain the depth map and cluster all pixels into two sorted clusters according to the depth values via k-means [15], which can be described as:

$$\{R_b, R_f\} = cluster(depth(I), 2), \tag{1}$$

where $R_b, R_f \in \{0,1\}^{W \times H}$ represents the background and foreground areas (as masks), respectively. The background area is further segmented into several disjoint regions according to the in-uniform geometry due to the presence of the foreground objects by the Meanshift algorithm [3], denoted as:

$$R_b = R_b^1 \cup R_b^2, ..., \cup R_b^n, \tag{2}$$

where $n$ denotes the number of background sub-regions and the sub-regions are sorted by area size. Brushstrokes within these sub-regions can be planned independently. For foreground areas that

are rich in semantics and texture, we further decompose it into instance-level semantic regions (e.g., a face, a human body, etc.), so that we can plan strokes individually for each semantic object region. For such a purpose, we utilize Automatic Masks Generator (AMG) from SAM model [10] to produce a series of object instance masks/regions (sorted by area size) as:

$$R_f = R_f^1 \cup R_f^2, ..., \cup R_f^m. \tag{3}$$

However, we observe that directly using AMG for foreground instance segmentation results in coarse-grained semantic regions. To obtain more fine-grained semantic regions (i.e., object parts), we adopt a prompt strategy guided by image gradients to SAM for further subdivision of the foreground masks into various textured regions which are highly probably corresponding to different object parts. Observing that areas with greater image gradients most likely correspond to regions that are richer in semantic information, we propose to guide SAM to retrieve more fine-grained foreground masks based on the spatial distributions of image gradients.

More concretely, we follow Im2Oil [20] to extract smoothed image gradient map $G$ using Sobel filter and Mean filter. For each foreground mask $R_f^i$, we firstly generate a masked gradient map $G_f^i$. Then, we sample one point with the greatest gradient in $G_f^i$ and use its circular neighborhood as a prompt for SAM to generate a fine-grained mask $R_f^{i,1}$. When the intersection over union (IoU) between $R_f^{i,1}$ and $R_f^i$ is less than a threshold $\epsilon_1$, we consider $R_f^{i,1}$ to be a meaningful subdivision. We retain $R_f^{i,1}$ and then remove the corresponding areas from $R_f^i$ and $G_f^i$. This process is recursively performed until the IoU between the newly generated mask and the remaining image area exceeds the threshold, as shown in Alg. 1. Finally, we obtain a set of fine-grained regions (sorted by area size) as:

$$R_f^i = R_f^{i,1} \cup R_f^{i,2}, ..., \cup R_f^{i,o_i}, \tag{4}$$

with

$$R_f^{i,j} = SAM(R_f^i \setminus \cup_{k=1}^{j-1} R_f^{i,k} \cdot I, \; sample(G_f^i \setminus \cup_{k=1}^{j-1} G_f^{i,k})), \quad (5)$$

where $sample(\cdot)$ denotes sampling a point with the greatest gradient value from the gradient map and generating its circular neighborhood. After subdividing all the foreground masks, we parse/arrange the original image into a tree of hierarchical semantic painting regions:

$$\{\{R_b^1, ..., R_b^n\}, \{\{R_f^{1,1}, ..., R_f^{1,o_1}\}, ..., \{R_f^{m,1}, ..., R_f^{m,o_m}\}\}\}, \quad (6)$$

and it is noted that all regions within each hierarchical level are arranged in descending order of area size.

---

**Algorithm 1:** Semantic Content Stratification

**Input** : $I, G, \epsilon_1$;
    // Input Image, Gradient Map, Threshold
$D = DAM(I)$; // Depth Map
$\{R_b, R_f\} = kmeans(D)$; // cluster by depth values
$\{R_b^1, R_b^2, ..., R_b^n\} = sorted(MeanShift(R_b))$;
    // cluster by pixel coordinates
$\{R_f^1, R_f^2, ..., R_f^m\} = sorted(SAM(R_f \cdot I, auto))$;
    // segment coarse foreground areas
**for** $R_f^i = R_f^1 : R_f^m$ **do**
    // segment fine foreground areas
    Initialize an empty set $T_f^i$ and a mask $R_{new} = \{0\}^{W \times H}$;
    **do**
        $R_f^i = R_f^i \setminus R_{new}$; $G_f^i = G \cdot R_f^i$;
        Find the point $p$ with greatest value in $G_f^i$;
        Generate a circular neighborhood $\mathcal{N}(p)$ for $p$;
        $R_{new} = SAM(R_f^i \cdot I, prompt = \mathcal{N}(p))$;
            // retrieve a fine mask
        $T_f^i.append(R_{new})$;
    **while** $IoU(R_{new}, R_f^i) \leq \epsilon_1$;
**end**
**Output** : Hierarchical regions $\{\{R_b^1, ..., R_b^n\}, \{T_f^1, ..., T_f^m\}\}$.

---

## 3.2 Coarse-Grained: Rough Structure Fill-in

After obtaining the above hierarchical painting semantic region parsing, we first draw a few number of rough strokes in each semantic area to fill-in the overall structure/abstract layout of the target painting. Mathematically, we follow previous methods [8, 14, 20] and utilize parametric rectangular stroke model as our painting primitive, where a stroke $s$ can be denoted as $\{x, y, w, h, \theta, r, g, b\}$ with $(x, y)$ representing the coordinates of the stroke rectangular center, $(w, h)$ representing the stroke size, $\theta$ denoting the angle of counter-clockwise rotation and $(r, g, b)$ meaning the template stroke color. Such a stroke can be painted on canvas by modifying its color and transferring its shape and location in canvas coordinate system through affine transformation as described in [14], which is a differentiable process.

For algorithmic simplicity, we allocate $N_b$ strokes for background regions and $N_f$ strokes for foreground regions considering that the foreground part has more complex textures. For each sub-region, we adaptively allocate the specific number of brush strokes based on its area size. More specific, we define the size factors for

background and foreground areas as $\sigma_b^i = size(R_b^i)/size(R_b)$ and $\sigma_f^{i,j} = size(R_f^{i,j})/size(R_f)$ respectively. Then the number of strokes needed for painting region $R_b^i$ and $R_f^{i,j}$ can be described as:

$$N_b^i = \lceil \sigma_b^i N_b \rceil, \; N_f^{i,j} = \lceil \sigma_f^{i,j} N_f \rceil, \quad (7)$$

where $\lceil \cdot \rceil$ denotes ceiling function.

Instead of learning based schemes, we adopt a gradient descent algorithm to directly optimize the parameters of the rough strokes for the sake of generalization and computational efficiency. We first initialize strokes by randomly sampling points within each painting region, and use their coordinates and colors as the initial positions (i.e., $\{x, y\}$) and colors (i.e., $\{r, g, b\}$). The stroke size is initialized as below:

$$[w_b^i, h_b^i] = \sigma_b^i [w_b, h_b], \; [w_f^{i,j}, h_f^{i,j}] = \sigma_f^{i,j} [w_f, h_f], \quad (8)$$

where we set $[w_b, h_b] = \frac{1}{\alpha}[W, H]$, $[w_f, h_f] = \frac{1}{8\alpha}[W, H]$ with $\alpha$ controlling the stroke sizes. Subsequently, we add all initial strokes onto a blank canvas in a hierarchical order from background to foreground and from large to small areas to create a new painting $I_c$. We then iteratively compare it with the input image $I$ and optimize all stroke parameters via gradient descent. In order to encourage the strokes to roughly reflect the semantic layout in the image while also achieving visual consistency, we follow [22] and utilize a pre-trained image encoder model of CLIP [17] to measure the semantic distances between the input image and the output painting. The overall loss function for optimizing is formulated as:

$$\mathcal{L} = \|I - I_c\|_2^2 + \beta dist(CLIP(I), CLIP(I_c)), \quad (9)$$

where $dist(\cdot, \cdot)$ is the cosine distance and $\beta$ is utilized to balance loss contributions.

## 3.3 Fine-Grained: Region Depiction via Multi-Scale Stroke Planning

For each semantic painting region, an artist typically begins filling with a rough sketch and then adds layers of details with brushes of smaller and smaller sizes. Therefore, we plan a specific number of painting scales/layers (here we plan 2 and 4 painting layers for the background and foreground regions respectively considering image texture distribution), and then search brushstrokes with progressively decreasing scales/granularities/sizes layer-by-layer, depicting the region/segment under consideration in a coarse-to-fine manner. The proposed multi-scale stroke search algorithm is shown in Alg. 2, which is elaborated as follows.

**Layer-by-layer optimization**: Starting from the coarsest/initial scale layer, the proposed layer-wise iteration scheme works the following way. At each scale, first we apply an annealing kernel $\mathcal{F}(\cdot, i)$ to obtain a smoothed version of the error map. Here, the kernel size anneals with the layer: $k^i = \lceil (\frac{1}{2})^{i-1} k^1 \rceil$, namely, smaller size kernels are used in finer-scale layers. The purpose of this pre-blurring operation is to guide the stroke searcher/planner to focus on only strong/high-contrast textured structures such as edges and ignoring the spurious image textures or noisy errors. The philosophy of using smaller blur kernels for later layers/finer scales is that as the stroke sizes we adopted decrease layer-by-layer, we believe low frequency errors have been mostly eliminated during previous iterations and thus we should force the stroke planner to

concentrate on higher frequency errors at finer-scale layers, corresponding to using small-scaled image filtering kernels. For strokes searched at each scale/layer, we set $w_{min}^i = \frac{1}{2}k^i$, $w_{max}^i = 4w_{min}^i$ as the minimum and maximum stroke widths limits, thus encouraging finer-grained strokes to reduce finer-scale painting errors. We set $h_{min}^i = 2w_{min}^i$, $h_{max}^i = 2w_{max}^i$ to prevent the search from producing strokes that are either too short or too long. Specific to each painting regions/segment, the minimum and maximum widths/lengths are further clamped to $\sigma w_{min}^i/\sigma h_{min}^i$ and $\sigma w_{max}^i/\sigma h_{max}^i$ respectively, where $\sigma$ is the size factor for each region, as defined in Sec. 3.2. The stopping criterion for terminating stroke searching in the current scale/layer and turning to the next layer search are set as: the IoU threshold $\epsilon_2$ of the layer-wise total strokes with respect to the masked region is less than 0.98 and the consecutive search failure number exceeds a threshold $c_f^1 = 5$ to avoid repetitive smearing of over-coarse sized strokes. The case of search failure is defined in the following paragraph.

**Within-layer optimization**: At each layer, to search each stroke parameter $\{x, y, w, h, \theta, r, g, b\}$, according to the pixel-wise error map $E_c$ between current canvas $I_c$ and the target image $I$, we propose to place the initial stroke position, $i.e.$, $(x_0, y_0)$ at points with large errors. Then, we use the color of the sampled point in the original image, $i.e.$, $(r_0, g_0, b_0) = I[x_0, y_0]$ as the color for this brushstroke. The next is to determine the rotation direction $\theta$ of the stroke. Although [20] achieves superior results with Edge Tangent Flow (ETF) determining stroke direction, it is only suitable for areas with clear gradient flows, such as hairs. For generality, we use image gradients extracted by Sobel filter as the search direction for brushstrokes. Based on the initial point $(x_0, y_0)$ with stroke color $(r_0, g_0, b_0)$ and direction $\theta$, we follow the common strategy [20] to search the stroke size $(w, h)$ by moving the search point $(x, y)$ pixel-by-pixel in four directions $\{\theta, \theta + \pi, \theta + \frac{\pi}{2}, \theta - \frac{\pi}{2}\}$. In addition to the commonly used two search termination conditions, $i.e.$, $|I[x, y] - I[x_0, y_0]| \leq t_c$ and $w_{min} \leq w \leq w_{max}, h_{min} \leq h \leq h_{max}$, we also add a new condition: the search point must not exceed the painting area. This restricts stroke search within each painting region, which not only greatly improving the efficiency of stroke search ($i.e.$, enabling parallel searching) but also ensuring precise alignment of brushstrokes with image structures such as edges and boundaries. After searching a stroke, we add it into current canvas and update the error map $E_c$. If the $L_2$ loss in the corresponding region decreases, we retain this stroke and then sample the starting point for the next search from the remaining points in this region. If the $L_2$ loss in the corresponding region increases, we consider this stroke fails and abandon it, and then resample a start point and search again in this region.

## 3.4 Edge Compensation Module

Our semantic content stratification module may result in formation of voids or overlaps at the boundaries of painting regions, leading to the ignorance of the masked error map to accurately capture this information. Moreover, the convolution of filters at the edges may reduce the sampling rate of edge pixels. Additionally, the rectangular strokes may partially cross beyond the boundaries of the painting regions. These three factors collectively diminish the representation of edge details. To address these issues, we develop an edge compensation module aiming at more detailed visual effects. Concretely, after completing the multi-scale stroke search, we calculate the error map between the current canvas and the target image, and use the image gradient field to weight the error map. Then, we iteratively sample starting points from the weighted error map to perform stroke search with the same constraints as the last layer in the progressive stroke search as described in Sec. 3.3.

Note that in the edge compensation module, we search stroke parameters in the global image and terminate the search process when the number of consecutive search failures exceeds a predefined threshold $c_f^2$.

---

**Algorithm 2:** Multi-scale Stroke Search

**Input** : $I, I_c, \mathcal{F}(\cdot, \cdot), R, \sigma_R, k^1, \epsilon_2, c_f^1$;
// Input Image, Current Canvas, Painting Region, Size Factor
// Filter, Initial Kernel Size, IoU and Failure Number Threshold

**Init:** $E_c = L_2(I, I_c) \cdot R$; // local error map
**for** $i=1:4$ **do**
$\quad k^i = \lceil (\frac{1}{2})^{i-1} k^1 \rceil$;
$\quad w_{min}^i = \frac{1}{2}k^i$; $w_{max}^i = 4w_{min}^i$;
$\quad h_{min}^i = 2w_{min}^i$; $h_{max}^i = 2w_{max}^i$;
$\quad n_{fail}^i = 0$; // number of consecutive search failures
$\quad R_{stroke}^i = \{0\}^{W \times H}$; // stroke areas
$\quad$ **do**
$\quad\quad (x_0, y_0) = sample(\mathcal{F}(E_c, i))$;
$\quad\quad$ Search a stroke $s$ starting from $(x_0, y_0)$ constrained
$\quad\quad\quad$ by $(\sigma_R w_{min}^i, \sigma_R w_{max}^i)$ and $(\sigma_R h_{min}^i, \sigma_R h_{max}^i)$;
$\quad\quad I_c' = draw(I_c, s)$; $E_c' = L_2(I, I_c') \cdot R$;
$\quad\quad$ **if** $mean(E_c') \leq mean(E_c)$ **then**
$\quad\quad\quad I_c = I_c'$; $E_c = E_c'$; $R_{stroke}^i = R_{stroke}^i \cup mask(s)$;
$\quad\quad\quad n_{fail}^i = 0$; search failure consecutiveness is broken
$\quad\quad$ **else**
$\quad\quad\quad I_c = I_c$; $E_c = E_c$; $n_{fail}^i += 1$;
$\quad\quad$ **end**
$\quad$ **while** $n_{fail}^i \leq c_f^1$ and $IoU(R_{stroke}^i, R) \leq \epsilon_2$;
**end**

---

## 4 EXPERIMENT

### 4.1 Datasets and Settings

**Datasets.** We adopt the gallery dataset from Im2Oil [20] and supplement it with five categories: scenery, portrait, animal, building, and still life, to form a dataset of one hundred images.

**Implementation details.** Our framework includes several hyper-parameters. In the semantic content stratification module, when the IoU of a newly retrieved mask with its corresponding region exceeds 0.8, we consider that region as no longer subject to finer-grained segmentation, $i.e.$, $\epsilon_1 = 0.8$. In the rough structure fill-in module, we set $N_b = 10$ and $N_f = 40$. In fact, $N_b$ and $N_f$ can be flexibly adjusted according to the texture distribution of the image's background and foreground regions. We set $\alpha = 8$ to initialize the size of the coarse brush strokes. $\beta$ is set to 1.0 to control loss contributions. During multi-scale stroke search, we use a gaussian filter with an initial size of $k^1 \times k^1$ with $k^1 = \lceil \frac{1}{32} min\{W, H\} \rceil$, where $[W, H]$

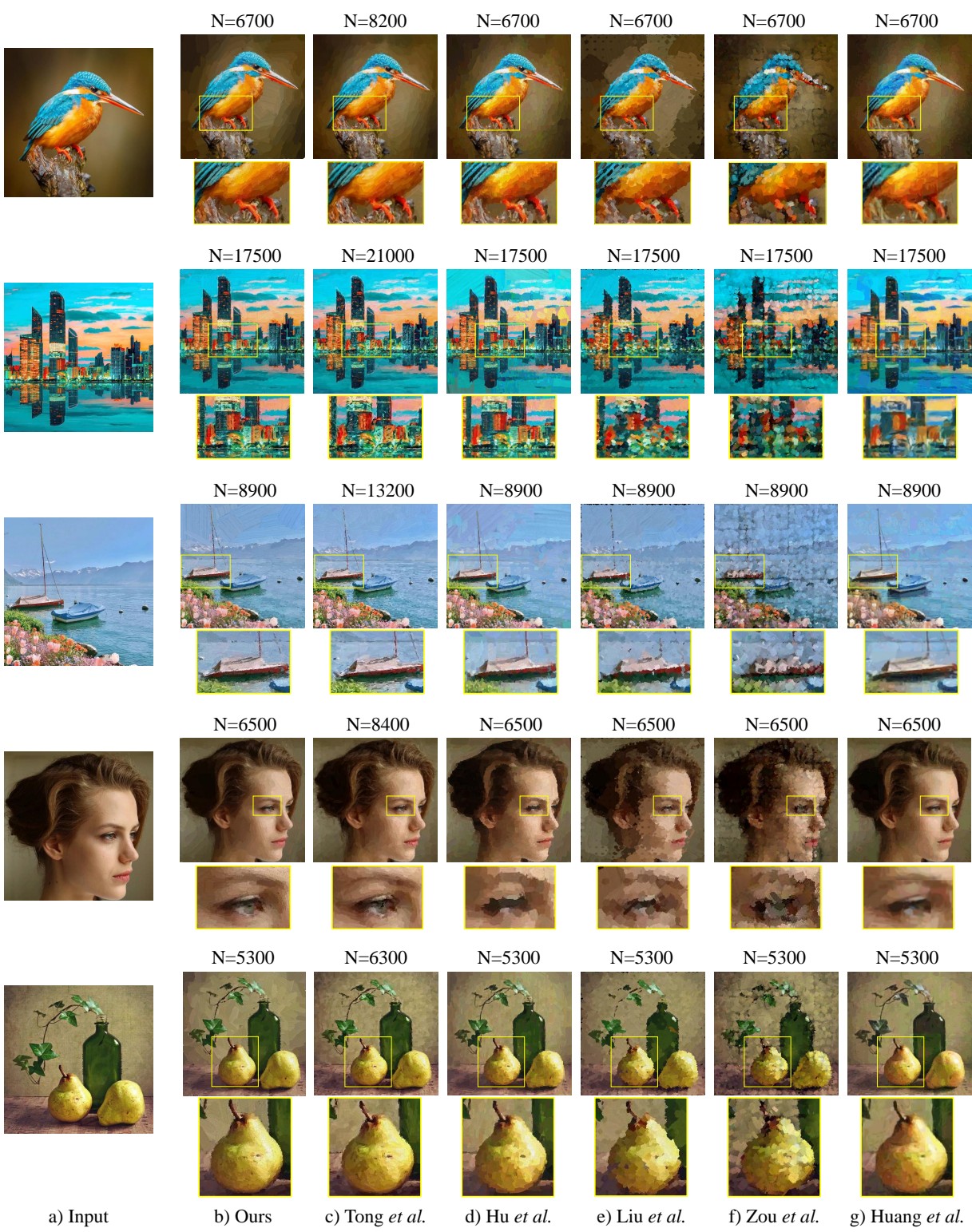

**Figure 3: Qualitative comparison with state-of-the-art methods. We showcase the rendering results of previous methods and ours. $N$ denotes the number of strokes used. As shown, our method achieves significantly better painting rendering effect than other methods. Im2Oil produces comparable rendering quality, but our method substantially reduces the number of strokes. Please zoom in to capture more details.**

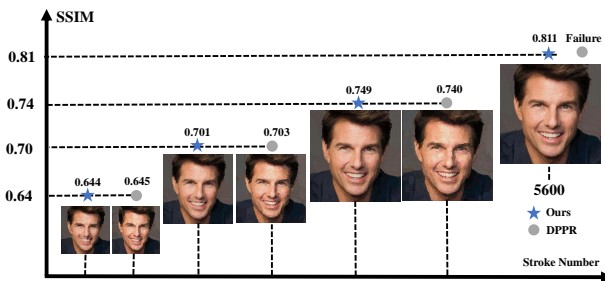

**Figure 4: We compare the stroke planning convergence speed with DPPR [8]. As shown, our method is able to the same painting quality as DPPR with significantly fewer strokes.**

denotes the image size. Regarding the termination condition of the stroke search, we set $t_c = 0.05$, $\epsilon_2 = 0.98$ and $c_f^1 = 5$. During edge compensation, since the search is based on the entire image, we terminate the search when the number of consecutive search failures exceeds 8, *i.e.*, $c_f^2 = 8$. All experiments are conducted on a 3090 GPU. The runtime it takes to paint an image is on the order of minutes due to the highly parallel stroke search.

## 4.2 Comparison with State-of-the-Art Methods

*4.2.1 Qualitative Comparison.* We compare our method with five state-of-the-art SBR methods, including Tong *et al.* [20], Hu *et al.* [8], Liu *et al.* [14], Huang *et al.* [9] and Zou *et al.* [25]. All compared methods are experimented with their default official parameters and for Tong *et al.* [20] we adopt the $p_{max} = \frac{1}{4}$ version. For Tong *et al.* [20], the number of strokes is determined by their search strategy. For other methods, we set the number of strokes the same as ours. The qualitative results are shown in Fig. 3.

We observe that our method achieves the best rendering results with the most realistic painting textures and the most intricate visual details. Previous methods produce significant stroke redundance on fine-grained areas like eyes due to the accumulation of small strokes. In contrast, our approach utilizes multi-scale strokes and encourages the use of larger strokes for initial filling, followed by finer-grained strokes to add details. Consequently, our method does not produce a piled-up effect and achieves higher-quality strokes with practical/artistic value. Huang *et al.* [9] and Liu *et al.* [14] all exhibit boundary discontinuities in their paintings. In contrast, our hierarchical painting areas naturally align strokes with the hierarchy of image structure/semantics, inherently avoiding edge discontinuities and conforming to artistic painting habits. Hu *et al.* [8] fail to capture the fine-grained details of images because their stroke parameters are predicted by neural networks, which often overlook features of fine-grained areas (with only a few pixels) such as human eyes and hairs. Tong *et al.* [20] produce visually appealing results, but suffer from issues like stroke overstacking and swirl-like errors at areas with complex textures. In contrast, our localized multi-scale stroke search strategy effectively reproduces fine-grained details. In addition, our method achieves better rendering quality while reducing the number of strokes by **20%** in average compared to Tong *et al.* [20], primarily due to the rough structure fill-in and efficient localized stroke search.

DPPR [8] is currently the state-of-the-art method that can quickly converge to the abstract structure of an image with few strokes. Thus we use SSIM as a metric to compare the convergence speed with DPPR, as shown in Fig. 4. When achieving the same painting quality (SSIM value), the number of strokes we need is significantly fewer than DPPR. When further increasing the stroke number, our method achieves significantly better painting results than DPPR.

*4.2.2 User Study.* To further demonstrate the superiority of our method, we invite 47 volunteers to evaluate the painting quality of five state-of-the-art methods and ours. Specifically, for each image in the dataset, we generate a group of six oil paintings with a shuffled order. Then volunteers are requested to observe each group of paintings for at least one minute, and provide a rating for each painting, with the score ranging from 1 to 6, where 1 means the best painting of this group. To prevent volunteers from scoring arbitrarily and leading to an unfair evaluation, we ask volunteers to distinguish the areas in each input image that could be prominently painted by human artists and pay special attention to these areas when scoring.

We calculate the average score for each method across each set of images. The average scores for different methods are presented in Tab. 1. We see that users show obvious preference to paintings produced by our method, which can be attributed to that our method greatly aligns the stroke with image semantics and depicts intricate and realistic textures in key areas of the image, such as human eyes and leaves.

**Table 1: User study results. Set 1-5 correspond to still life, animal, portrait, scenery, and building, respectively.**

| Methods | Set-1 | Set-2 | Set-3 | Set-4 | Set-5 | Mean |
|---|---|---|---|---|---|---|
| Huang *et al.* [9] | 4.88 | 4.87 | 2.55 | 4.74 | 4.91 | 4.39 |
| Liu *et al.* [14] | 5.11 | 5.02 | 5.98 | 4.87 | 5.11 | 5.22 |
| Zou et.al. [25] | 5.01 | 5.11 | 5.17 | 5.38 | 4.98 | 5.13 |
| Hu *et al.* [25] | 2.38 | 1.83 | 4.85 | 2.09 | 2.34 | 2.70 |
| Tong *et al.* [20] | 2.09 | 2.36 | 1.83 | 2.42 | 2.28 | 2.20 |
| Ours | **1.55** | **1.81** | **1.62** | **1.49** | **1.38** | **1.57** |

*4.2.3 Stroke Quality Assessment.* Current SBR methods commonly use pixel loss to evaluate the generated strokes, lacking a direct quality assessment for stroke sequence.

Our belief is that high-quality stroke sequence should allow users to understand stroke parameters from a semantic perspective and provide interactive suggestions modifications. Therefore, we integrate the SBR algorithms into an interactive simulation-based painting system, using the generated stroke sequence to drive a physical brush to draw realistic painting pigments onto the canvas for rendering and then evaluate the quality of the painted results, which reflects the degree of alignment between generated strokes and realistic painting strokes.

As for a stoke $\{x, y, w, h, \theta, r, g, b\}$, we can easily calculate the start and end points of the painting stroke in the $h$-direction, while $w$-direction represents the brush width. By using linear interpolation, we can obtain the coordinates of the brushstroke movement trajectory, so as to drive the brush to move on canvas according to the generated stroke parameters.

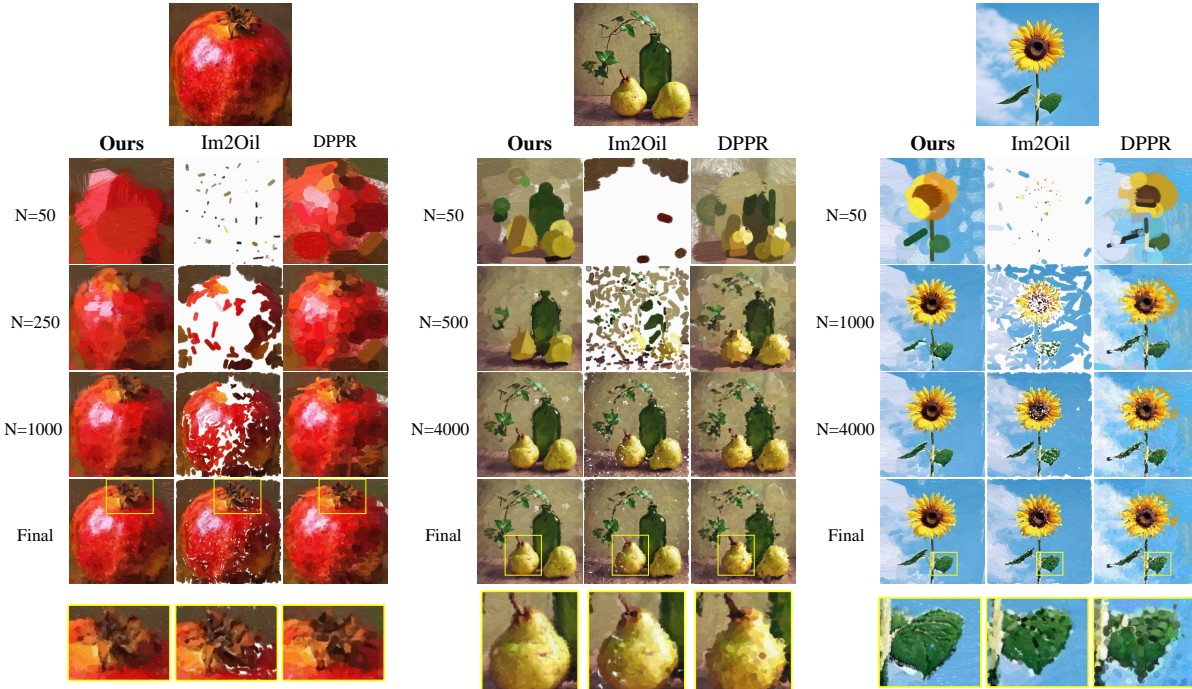

**Figure 5: Painting results produced by simulation-based painting system. We also show the painting process of the painting system. Our method generates high-quality strokes that are suitable for practical automatic painting agents.**

We compare stroke quality with two methods with great rendering effects (DPPR [8] and Im2Oil [20]). The results are shown in Fig. 5. We see that the simulated painting results of DPPR [8] are very poor(*e.g.* the sunflower image). The reason is that DPPR truncates the strokes using a rectangular painting region before rendering them onto the image during training, hence the stroke parameters do not align with the visual information. However, in the real painting process, it is not feasible to truncate each stroke with a corresponding rectangle, making the strokes generated by DPPR lack practical value. Im2Oil [20] suffers from highly disorganized stroke order which does not align with human painting habits. Additionally, it fails to adequately display fine-grained details. Compared to these two methods, our method plans the stroke sequence in an artist-like fashion, which better aligns with real painting process.

## 4.3 Ablation Study

We evaluate the impact of some important components in our method by individually removing them. We use the set-3 (portrait) in the dataset to compare the image quality and number of strokes with each module being removed individually. Tab. 2 shows the comparison results. **(1)** Removing semantic content stratification results in no alignment between semantics and strokes, leading to repeatedly smudging in local areas. Additionally, the program can only search for strokes serially, resulting in unacceptable time consumption. **(2)** Removing rough structure fill-in module makes the stroke number increase while maintaining rendering quality. **(3)** Removing filtering mechanism leads to that fine textures are easily captured at coarse level strokes searching and then are repeatedly

smeared, increasing the number of strokes. **(4)** Removing edge compensation results in unacceptable artifacts in region boundaries. **(5)** We also compare an ablated verion without $\mathcal{L}_{CLIP}$ in rough structure fill-in, which makes the convergence speed decrease and the strokes tend to span across multiple semantic areas, leading to an increasing number of strokes at following edge compensation module.

**Table 2: Ablation study on key components of our method. This experiment is conducted on the portrait dataset.**

| Module | $L_2$ Dist ↓ | PSNR ↑ | Stroke number |
|---|---|---|---|
| w/o semantic stratification | 0.0022 | 26.49 | +38.4% |
| w/o rough structure fill-in | 0.0016 | 27.96 | +17.3% |
| w/o filtering mechanism | 0.0017 | 27.74 | +28.9% |
| w/o edge compensation | 0.0020 | 26.89 | -4.5% |
| w/o $\mathcal{L}_{CLIP}$ | 0.0016 | 28.02 | +6.4% |
| Ours | 0.0015 | 28.15 | - |

## 5 CONCLUSION

In this paper, we present a novel SBR method which aligns with artistic painting techniques through decomposing the input image into hierarchical painting regions and generating multi-granularity strokes in a two-stage manner. We also integrate SBR frameworks into a simulation-based painting system for stroke quality assessment. Extensive experimental results show that our method achieves high fidelity rendering effects with a reduced number of strokes and exhibits great stroke quality over previous methods.

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
