# OpenReview forum: "Towards Artist-Like Painting Agents with Multi-Granularity Semantic Alignment"
_acmmm.org/ACMMM/2024/Conference — MM2024 Poster_

### Official Review · Reviewer_ydor · 2024-05-06

**Rating:** 4
**Confidence:** 2

**Summary:**

This paper proposes a new stroke-based rendering (SBR) method which enforces the generated sequence of strokes to align with artistic oil painting styles. To achieve this goal, this paper first introduces a semantic content stratification module to decompose the input image into hierarchical painting regions, and then a two-stage artist-like stroke generation/planning module along with an edge compensation module is designed to generate multi-granularity strokes.

**Strengths:**

1. The idea of incorporating artist experience/knowledge into artistic-like painting generation is interesting and reasonable.
2. This paper is well-motivated and well-organized.
3. Extensive experiments have been conducted to support their claims.

**Limitations:**

1. This paper is similar to Intelli-Paint [19] in many aspects. Intelli-Paint also introduces a progressive layering approach, which much like a human, allows the painting agent to draw a given scene in multiple successive layers. Intelli-Paint proposes a brushstroke regularization strategy which allows for 60–80% reduction in the total number of required brushstrokes. Despite the highly related relationship, this paper did not compare the proposed method with Intelli-Paint.

2. From my perspective, the visual quality of the results generated by the proposed method is not satisfying. The images generated by the proposed method are more like photographs instead of paintings.

3. The ablation study didn’t provide qualitative results (neither in the main paper nor in the supplementary material) which are the most direct and clear way to reflect the effect of each component.

**Suitability:**

2

---

### Official Review · Reviewer_CeBY · 2024-05-23

**Rating:** 3
**Confidence:** 3

**Summary:**

The paper introduces an innovative image-to-painting framework that aims to replicate an artist's techniques. This framework divides the target image into painting regions with different semantic levels and employs a coarse-to-fine brushstroke sequence for parallel rendering. As a result, it achieves efficient and high-quality artistic effects. This method not only generates high-fidelity paintings with fewer brushstrokes but also better captures the details of the image.

**Strengths:**

1. The hierarchical painting approach employed in the paper exhibits a remarkable ease of comprehension and seems very easy to follow.
2. The qualitative visual results presented in the paper serve as a testament to the efficacy of the proposed method, vividly showcasing the intricate details of the rendering process.

**Limitations:**

After a thorough examination of the article, I still have some queries that I hope you can address:

1. The paper exclusively contrasts the SSIM metric with DPPR, omitting a more thorough and objective quantitative analysis. It would greatly benefit readers to include a broader spectrum of objective evaluation metrics to validate the efficacy of the proposed method.

2. The clarity and readability of the writing could be enhanced. Specifically, I found it challenging to comprehend the "existing issues" referenced in the Abstract, with no subsequent elucidation provided in the text. Additionally, the contributions delineated in the Introduction seem somewhat ambiguous. Moreover, regarding the approach outlined in Section 4.2.2 concerning the User Study aimed at preventing arbitrary scoring by volunteers and mitigating potential biases in evaluation, I feel that further clarification is needed. Providing additional explanations would undoubtedly facilitate readers' comprehension of this study.

Thank you for considering these suggestions.

**Suitability:**

3

---

### Official Review · Reviewer_PJtn · 2024-05-26

**Rating:** 5
**Confidence:** 2

**Summary:**

This paper presents a novel Stroke-Based Rendering (SBR) framework designed to align generated paintings with artistic oil painting techniques. The proposed method involves a semantic content stratification module that decomposes images into hierarchical painting regions, followed by a two-stage stroke generation process. Extensive experimental results demonstrate the method's ability to produce high-fidelity, artist-like paintings with a reduced number of strokes.

**Strengths:**

1. Innovative Approach: The hierarchical decomposition and coarse-to-fine stroke generation strategy is a significant improvement over existing methods. An edge compensation module to refine boundaries between painting regions and integration with a fluid simulation painting system for stroke quality assessment.

2. Comprehensive Evaluation: Extensive experiments, including qualitative comparisons and user studies, provide strong evidence of the method's effectiveness.

**Limitations:**

1. The paper does not provide an analysis of inference time, making it hard to assess its efficiency and practical use compared to related works like Intelli-Paint.
2. The paper does not discuss failure cases, which is a crucial aspect of understanding the method's limitations.
3. The method heavily relies on external models, and its effectiveness is closely tied to the performance of these models, leading to variability and potential limitations in different application contexts.
4. Considering the complexity of the proposed methods, it is unclear if they can generalize well to all popular styles.


--------------
PS: will the code and model weights be released?

**Suitability:**

2

---

### Official Review · Reviewer_CtqC · 2024-06-09

**Rating:** 5
**Confidence:** 3

**Summary:**

This paper proposes a novel stroke-based rendering algorithm with multi-granularity semantic alignment, which aligns the stroke generation process with artistic oil painting styles, particularly tailored for artist-like painting agents.

**Strengths:**

1. The proposed framework can paint with fewer stroke numbers to obtain higher-quality painting results, compared to previous methods.
2. The user study and the various comparisons demonstrate the superiority of the proposed method.

**Limitations:**

1. What is the inference time, compared with previous methods? If it is too large than previous methods, it may not be a fair comparison.
2. Any failure case to show? The failure cases and limitations should be discussed in more detail. For example, one limitation may be that the whole method depends on accurate semantic segmentation results at the first stage. What if the segmentation is not high quality? How do they influence the stroke number or the quality of the results?

**Suitability:**

2

---

### Meta-Review · Area_Chair_hqPH · 2024-07-01

**Recommendation:** Accept (Poster)
**Confidence:** 5

**Metareview:**

This paper was reviewed by four experts in the field. The paper received mixed reviews WR, BA, WA, WA. The reviewers liked the idea of this paper, its thorough evaluation and good presentation. The reviewers also raised the following concerns on the application scenarios, and comparisons with other baselines. Based on the rebuttal, the AC feels that the additional results in the final version of the paper solve the concerns. Based on the reviews, the AC would like to recommend the acceptance of this paper and suggest the authors to include the added experiments in the rebuttal to the final version.